# Window-Based Multi-Objective Optimization for Dynamic Patient Scheduling with Problem-Specific Operators

**Ali Nader Mahmed and M. N. M. Kahar \***

Faculty of Computing, College of Computing and Applied Sciences, Universiti Malaysia Pahang,
Kuantan 26600, Pahang, Malaysia; alisheikhly@gmail.com
\* Correspondence: mnizam@ump.edu.my

**Abstract:** The problem of patient admission scheduling (PAS) is a nondeterministic polynomial time (NP)-hard combinatorial optimization problem with numerous constraints. Researchers have divided the constraints of this problem into hard (i.e., feasible solution) and soft constraints (i.e., quality solution). The majority of research has dealt with PAS using integer linear programming (ILP) and single objective meta-heuristic searching-based approaches. ILP-based approaches carry high computational demand and the risk of non-feasibility for a large dataset. In a single objective optimization, there is a risk of local minima due to the non-convexity of the problem. In this article, we present the first pareto front-based optimization for PAS using set of meta-heuristic approaches. We selected four multi-objective optimization methods. Problem-specific operators were developed for each of them. Next, we compared them with single objective optimization approaches, namely, simulated annealing and particle swarm optimization. In addition, this article also deals with the dynamical aspect of this problem by comparing historical window-based decomposition with day decomposition, as has previously been proposed in the literature. An evaluation of the models proposed in the article and comparison with traditional models reveals the superiority of our proposed multi-objective optimization with window incorporation in terms of optimality.

**Keywords:** patient admission scheduling (PAS); dynamic optimization; multi-objective optimization; non-dominated sorting genetic algorithm (NSGA-II); NSGA-III

## 1. Introduction

A management system involving resource allocation and patient scheduling is an essential element of high-quality healthcare services. Patient scheduling is crucial due to increasing pressure on services, especially during the COVID-19 pandemic. Patient scheduling requires consideration of strategic and other criteria before assigning rooms (or departments) to patients. These criteria might include patients' specific requirements, the necessary medical expertise needed for their treatment, room needs in terms of equipment, and other patient treatment preferences to help facilitate a speedy recovery. It is infeasible to consider such elements using a manual scheduling process [1]. Hence, several algorithms [2,3] were proposed in the last decade to handle these operational aspects when assigning beds to patients, while attempting to satisfy as many patient preferences as possible, considering critical medical resource availability. This problem is known as patient admission scheduling (PAS).

Optimization algorithms are used to improve the performance of systems in various domains and scopes, e.g., automatic voltage regulation [4] and parameters optimization [5]. Our literature review concerning the developed PAS algorithms indicates that the general direction for formulating PAS systems is to use mixed integer-based linear programming (MILP) problems with hard and soft constraints [2]. Other methods use meta-heuristic searching-based optimization, in which soft constraints are embedded in the objective function as a weighted average term, where penalties define the weights. The penalties are logical

assumptions concerning constraint importance assigned using weights; they have no practical interpretation when defining the real-world problem [6]. In addition, penalties assume problem convexity, i.e., the optimal point concerning the linear combination of soft constraints is the same as the optimal points being selected individually. This is not accurate due to the non-convex nature of the optimization surface. Hence, using non-dominated sorting concerning the soft constraints and individual objectives is more effective in providing non-dominated solutions and offering flexibility or a set of choices to the decision-maker. A third direction is to use machine learning approaches based on reinforcement learning, where the decisions-states are used to train single/multi-agent systems, with the aim of reaching optimal policy [7]. The difference between meta-heuristic searching and reinforcement learning is that the latter stores knowledge of state-decision rewards and uses this to update knowledge in an incremental way. The formulation of state, action or decision, and reward is a key factor in such performance optimization. On the other hand, meta-heuristic searching has a less incremental nature but offers a simpler solution.

This article proposes a novel multi-objective optimization to solve the problem of the PAS variant. We selected four methods: multi-objective particle swarm optimization (MOPSO), objective decomposition PSO (ODMPSO), non-dominated sorting genetic optimization (NSGA-II) and (NSGA-III). Problem-specific operators were developed for each of them. The remainder of this work is organized as follows: Section 2 lists the contributions. Next, Section 3 presents the findings of the literature review. Then, Section 4 provides an overview of the benchmarking algorithms. After this, the research methodology is given in Section 5. Section 6 then discusses the experimental work and results. Lastly, Section 7 presents the conclusion and guidance for future works.

## 2. Contribution

The work provides a novel approach for solving the patient admission scheduling (PAS) problem using a multi-objective optimization approach. This work makes several contributions:

(1) It provides a novel formulation of the dynamic PAS problem using the multi-objective optimization approach.

(2) The work updates the day-specific optimization concept with window-based optimization. The adaptive window is enabled. Its length is changed according to the number of newly arrived patients.

(3) The study design develops a novel multi-objective optimization algorithm for solving dynamic PAS by creating application-oriented operators and proposing a repository for saving non-dominated solutions.

## 3. Literature Survey

The literature consists of two sub-sections. Section 3.1 describes the literature specific to patient admission scheduling, while Section 3.2 discusses the use of multi-objective optimization algorithms for scheduling problems.

### 3.1. Patient Admission Scheduling

Patient Admission Scheduling (PAS) is an NP-hard problem [8]. Ref. [6] addressed the PAS problem using a combinatorial formulation of Integer Linear Programming (ILP) and the Tabu search algorithm. They aimed to determine patients' optimal bed assignment schedule using prior knowledge concerning hospital departments, room capacity, bed availability, specialties, equipment, and qualitative aspects, including patient preference based on gender, age, and room matching. The method provided good results with relatively small and medium data sizes. However, this work is challenged from several perspectives: an offline approach is impractical due to the dynamic aspects of the problem; taking a weighted average of the soft constraints can result in sub-optimal outcomes due to model non-convexity; this method presents only a single optimal solution, resulting in limited choices available to the decision-maker [9]. Other researchers [8] presented two Mixed Integer Programming (MIP)-based heuristic approaches, namely Fix-and-Relax

(F&R) and Fix-and-Optimize (F&O), where PAS problem instances are decomposed into sub-problems, which are then optimized. Specifically, the solutions generated by the F&R heuristic are used as inputs for the F&O heuristic to optimize the solutions. Patient Length-of-Stay (LOS), room preference, admission date, specialty requirement, and age are the factors considered for problem and time decompositions considering different optimization window sizes. Another team of researchers [10] proposed a different formulation extending the study by [6]; they used the dynamic patient-to-room assignment problem, which helped reduce the number of decision variables, lessened the computation of different values of lower bounds by omitting some constraints, and adapted simulated annealing to find the best solution. The study by [6] was improved by [11] using two levels of heuristics or hyper-heuristics comprising the addition of local search moves. This work used the great deluge algorithm for hyper-heuristics. It was found that the hyper-heuristic approach to the patient admission scheduling problem significantly outperforms the Tabu search approach. However, this research was criticized by [12] for the limitations of its deluge algorithm's linear decay rate, which was improved to a non-linear adaptive decay rate using the same soft and hard constraints [6]. The results have shown the importance of adaptive parameters for meta-searching. In the work of [13], the scheduling objectives were arranged as short- and long-term objectives and periodic re-optimization was used. Lower bounds were calculated using Dantzig–Wolfe decomposition and column generation. The results showed that the proposed short-term strategy produced long-term solutions of significantly better quality than the best-known strategy for 26 out of the 30 instances. Researchers [14] created a scheduling algorithm for planning medical tourist travel to Destination Medical Centers (DMCs). The objectives were to minimize patient delay, preferred starting day, and patients flow time. They used flow-shop scheduling and implemented optimization using Tabu search, simulated annealing, and simulations. The process is based on discrete event simulation that evaluates the solution, considering admission day, admission time, and patient sequence as decision variables over a one-day interval. The results show that average patient flow time increases significantly when patients arrive in batches, rather than every patient having a specific start time every day. In the work of [15], the existing deterministic model developed by [6] was modified to incorporate stochastic aspects. They used the Poisson distribution and discrete phase-type distribution to model arrivals and departures, respectively. Hence, their model became a stochastic variant of the older deterministic model. Appointment time modelling based on patient requirements and doctor performance (i.e., speed factor) was proposed. Their model is implemented using a solver for small scale problems and the genetic algorithm for more significant problems. However, it ignores multi-objective handling from the non-dominated sorting perspective.

Overall, the PAS problem has been addressed in the literature from various perspectives, and the solutions presented offer various levels of practicality in terms of adding soft constraints, LoS uncertainty, or acceptance of urgent cases. However, none of the mentioned approaches addressed the non-domination aspect of the solution. Specifically, the PAS problem is a multi-objective optimization problem when dealing with soft constraints as separate objectives. In this way, the decision-maker can be offered a flexible choice concerning an exact non-dominated solution to scheduling instead of the weighted average of soft constraints. The latter approach has risks relating to local minima due to non-convexity.

### 3.2. Multi-Objective Optimisation for Scheduling

Multi-objective optimization-based scheduling has received considerable attention in the recent literature. Multi-objective particle swarm optimization has been used for solving various scheduling problems and other applications. In [16], researchers proposed a modified multiple-objective particle swarm optimization (MMOPSO) to solve a mixed-integer mathematical programming model for the response phase of an earthquake. Modified multi-objective particle swarm optimization includes two local search operations. The model considers two objective functions: to minimize the total location cost and allocation of facilities and the amount of relief supplies shortage. This approach was superior to the

two famous non-dominated sorting genetic algorithms NSGA-II and the epsilon constraint method. In [17], the problem of container-based scheduling for Internet-of-Things in a cloud environment was optimized using a modified variant of multi-objective particle swarm optimization. Researchers considered two optimization objectives, namely, energy consumption and computational time optimization. Multi-objective aspects are dealt with using a weighted sum approach-based fitness function to evaluate the quality of the solution. The multi-objective particle swarm optimization was modified concerning convergence speed using an acceleration component. In addition, standard PSO uses current global best and individual best particles to find an optimal solution. The results show that their method performs better than existing methods in various cloud performance metrics. In another work [18], due to convergence speed and accuracy limitations, the PSO algorithm was modified based on velocity and displacement and named as the acceleration PSO (APSO) technique. It uses the global best at the individual level to facilitate convergence and reduce randomness for future iterations. Another modification of the particle swarm optimization for improving search performance was performed by [19], where the neighborhood of every particle was constructed. The optimal neighborhood solution was chosen using the self-organizing mapping (SOM) method. The results can provide the theoretical basis and concrete scheme reference for reservoir operation. In [20], an analytical investigation of the convergence of self-adaptive PSO (SAPSO) was performed by providing a parameter selection principle that guarantees convergence toward good quality solutions. SAPSO was leveraged to create the MOO framework, named SAMOPSO. To gain a well-distributed Pareto front, researchers also designed an external repository to store non-dominated solutions. Next, a circular sorting method was designed by integrating the elitist-preserving approach to update the external repository in the developed MOO framework. In [21], researchers adapted the particle swarm optimization method to solve high-dimensional discrete variable problems. They integrated PSO with variable neighborhood search and stretching techniques. The results obtained after solving high-dimensional issues were promising. Modification of the search algorithm is not restricted to particle swarm optimization. In [22], researchers modified the non-dominated sorting genetic algorithm to address the issue of surgery scheduling in operating rooms. The solved model is considered a resource allocation model, which focuses primarily on allocating ORs for each surgical specialty (SS). The modification of NSGA-II was performed in two parts: initializing population selection using a tournament technique. The results illustrate that the model can provide hospital managers with a series of "optimal" solutions to effectively allocate relevant resources and ORs for surgeries. Moreover, the authors show that the improved NSGA-II has high computational efficiency and is more suitable for solving large-scale problems. In [23], a multi-parent crossover genetic algorithm was proposed. The multi-parent concept defines a cross operator with n-string division points when working for n parents. Experimental results demonstrated that the multi-parent order crossover (MPOX) significantly improves order crossover (OX) in both problem domains. It outperforms both adjacency-based crossover (ABC) and multi-parent partially mapped crossover (MPPMX) over the travelling salesman problem and the berth allocation problem, and in less computational time. These results indicate the effectiveness of MPOX over OX, ABC, and crossover (MPPMX), and its capability for solving both problems.

Overall, multi-objective meta-heuristic search algorithms were found to be effective for solving scheduling applications with a multi-objective nature. Nevertheless, most approaches used these algorithms to solve problems with a small number of objectives. Converting PAS to a multi-objective problem entails a high number of objectives generated from the soft constraints. The inclusion of a high number of objectives requires special modification of the search criteria to assure smooth convergence behavior. In addition, we see that meta-heuristic multi-objective optimization algorithms, including particle- and genetic-based searching, have been used for scheduling applications. Furthermore, most of them cannot be applied directly, but require a specific operator design tailored to the nature of each application.

## 4. Background

In this section, we review four popular benchmarking algorithms for multi-objective optimization. The first is the non-dominated sorting genetic algorithm (NSGA-II), which is presented in Section 4.1. The second is NAGA-III, which is presented in Section 4.2. The third and fourth are multi-objective particle swarm optimization (MOPSO) and MOPSO with objective decomposition (ODMOPSO), presented together in Section 4.3.

### 4.1. Non-Dominated Sorting Genetic Algorithm NSGA-II

The non-dominated sorting genetic algorithm (NSGA-II) that is proposed by [24] operates on multiple objectives. The algorithm proposed a new concept called crowding distance (CA) to select a solution subset with the same rank across generations. Higher priority is given to solutions with higher crowding distance, making the algorithm more capable of exploring new areas in the solution space. The CA selection technique is applied after non-dominated sorting. The pseudocode of the algorithm is provided in Algorithm 1.

---

**Algorithm 1.** Pseudocode of non-dominated sorting genetic algorithm.

---

**Input**
Size of population
maximum generation
**Output**
Pareto front
**Start**
1—Initialize population
2—Evaluate the individual fitness
3—Select ranked individuals
4—**while** maximum generation is not reached
5— **while** non-offspring reached
6—　　Select parents
7—　　Crossover or mutation
8—　　Evaluate offspring
9— **end**
10— Select new generation using ranking and CA
11— **end**
12—Output = Pareto front
**End.**

---

### 4.2. Non-Dominated Sorting Genetic Algorithm III (NSGA-III)

It is important to distinguish between multiple-objective and many-objective algorithms. The first involves solving optimization problems with a limited number of objectives, such as two, three, or four. However, the latter involves solving problems with numerous objectives. There are various challenges involved in solving problems that include many objectives. First, numerous objectives cause search convergence challenges. Furthermore, searching becomes similar to a random search. Second, numerous objectives require a large number of solutions to converge. According to [25], problems with 4, 5, and 7 objectives require 62,500, 1,953,125, and 1,708,984,375 non-dominated solutions, respectively. Third, the high number of non-dominated solutions implies complex criteria calculations in order to determine the search direction. Fourth, the dissimilarity between parents and offspring makes recombination similar to random generation. The literature highlights a need to devise more effective search algorithms. In [26], researchers proposed a reference point-based framework that emphasizes non-dominated solutions closer to the reference points to maintain search diversity.

We present the pseudocode for selecting solutions based on the concept of reference points of NSGA-III in Algorithm 2. In [27], this algorithm is incorporated for the particle swarm optimization framework.

---

**Algorithm 2.** Pseudocode of the process of selecting non-dominated solutions based on the process of NSGA-III.

---

**Input**:
H structured reference points Zs or supplied aspiration
points Za,
parent population Pt
**Output**:
P(t + 1)
**Start**
1—St = Ø, I = 1
2—Qt = Recombination + Mutation (Pt)
3—Rt = Pt ∪ Qt
4—(F1, F2, . . . ) = Non-dominated-sort (Rt)
5—**repeat**
6—    (St = St ∪ Fi and i = i + 1
7—until |St| ≥ N)
8—Last front to be included: Fl = Fi
9— **if** |St| = N    then
10—      P(t + 1) = St, break;
11—**else**
12— P = all previous fronts
13— Points to be chosen from Fl: K = N − |Pt + 1|
14— Normalize objectives and create reference set Zr:
          Normalize(fn, St, Zr, Zs, Za)
15— Associate each member s of St with a reference point:
          [π(s), d(s)] = Associate(St, Zr) % π(s): closest
          reference point, d: distance between s and π(s)
16— Compute niche count of reference point
17— Choose K members one at a time from Fl to construct
          P(t + 1): Niching(K, ρj, π, d, Zr, Fl, P(t + 1))
18—**end if**
**End.**

---

*4.3. Multi-Objective Particle Swarm Optimization MOPSO and MOPSO with Objective DeComposition (ODMOPSO)*

This algorithm is a multi-objective variant of the particle swarm optimization technique. It uses non-dominated sorting for evaluating solutions using ranks, and it keeps a repository of non-dominated solutions. The repository is used to select one solution and move other solutions closer, as specified in mobility Equation (1).

$$VEL[i] = W \times VEL[i] + R_1 \times (PBESTS[i] - POP[i]) + R_2 \times (REP[h] - POP[i] \quad (1)$$

where *h* is an index generated by the repository using a probabilistic model. Generating this value for Equation (1) requires providing equal probability to single solutions located in a hypercube cell. In contrast, the probability of multiple solutions in the hyper-cube is divided based on the number of solutions in the hyper-cube, which lowers the probability so as to prioritize vacant areas in space thereby pulling the solutions and creating more diversity. The pseudocode is presented in Algorithm 3.

---

**Algorithm 3.** Pseudocode of multi-objective particle swarm optimization.

---

**Input**
Max //maximum number of particles
definition of objective functions
repository size
granularity of hyper-cube
**Output**
pareto front
**Start**
1—generate the first swarm
2—initiate the speed of the particles
3—evaluate each of the particles
4—store non-dominated particles in the repository
5—generate hyper-cube using the granularity of hyper-cube
6—initialize the memory of each particle
7— **while** the maximum number of iterations is not reached
8—　move the particle using selected solution from the repository as best global
9—　maintain solutions that exceed the boundary so they lie in the valid region of space
10—　evaluate the solutions using the objective functions
11—　update the repository
12— **end**
**End.**

---

## 5. Methodology

The methodology includes two developments: the first relates to genetic optimization operators, and the second relates to particle swarm optimization operators.

We present crossover and mutation for the former and the movement operator for the latter. Both share the same initialization process.

### 5.1. Problem Formulation

Assuming that we have a hospital consisting of a set of rooms $R_j, j = 1 \ldots R$, the room in the hospital is denoted by $R_j$ and each room contains a number of beds, where each bed $B$ in room $R_j$ is denoted as $B_{jB}$. We assume that patients remain in hospital for a number of nights $N_k, k = 1 \ldots T$. The hospital contains the set of departments $D_m, m = 1 \ldots D$ and each department supports the set of specialisms $S_l, l = 1 \ldots S$. The goal is to develop a schedular that assigns to each patient a planning period and a room within its planning period. The room should be available within the planning period. The length of stay is contiguous. Two patients should not be assigned the same bed in the same night. Other requirements for patients have to be fulfilled, namely, assigning a room that supports the specialism required for their treatment, the need for some patients to be allocated a single room, and allocation to the department suited to the age of the patient. After assuring that the previous requirements are fulfilled, the schedular aims to optimize the following objectives:

1—maximum respect of the patient's room choice (single, twin or ward);
2—maximum support from the department to the patient's disease;
3—maximum support from the room to the patient's disease;
4—minimum unplanned transfer of patients.

### 5.2. General Block Diagram

Figure 1 shows the block diagram for performing window-based patient scheduling using multi-objective optimization. The block diagram's input is the new patients who have arrived at the hospital. They are placed inside the window alongside the non-confirmed patients from the previous optimization execution. Patients who are planned for later than D days are defined as non-confirmed; we think they are non-confirmed because they are allowed to be rescheduled as long as the scheduling is later than D days. The coordinator

is the next component, and it is in charge of pulling patients from the window and feeding them into the multi-objective optimization input.

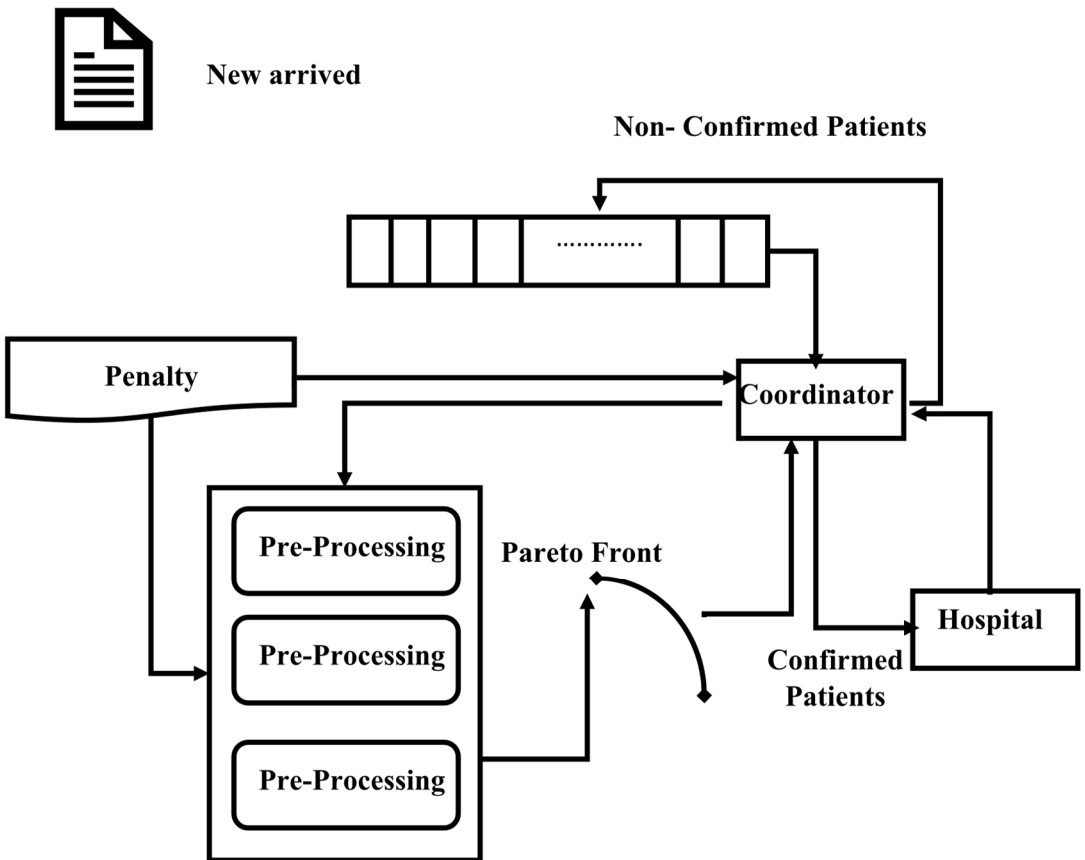

**Figure 1.** Block diagram for performing window-based patient scheduling using multi-objective optimization.

### 5.2.1. Initialization Algorithm

The initialization algorithm is responsible for generating the initial solution within the window, which represents the number of days that accommodates a certain number of new patients. As it is given in Algorithm 4, the inputs of this algorithm are: $S_{pre}$, which denotes the solution found using the previous window, and *Data*, which denotes the data concerning the list of rooms, updated patient lists, and patient-room suitability. The output is $S_{current}$, which denotes the solution after optimization based on the current window and updated patient list. The algorithm iterates on List-new-patients and initiates a variable named Room with the value of $-1$, indicating that a suitable room has not been found yet for this patient. Patients are associated with one of two cases: delayed or not delayed.

In the former case, the algorithm checks the room from the previous solution to determine suitability. The patient is assigned the room if it is suitable and vacant. Otherwise, the algorithm selects a random room for the patient if rooms are available. If additional rooms are unavailable, the patient is assigned a room from the previous solution or a random room, and delay is indicated using value 1 for the delay flag.

| **Algorithm 4.** The generation of the initial solution. |
| --- |

**Input**:
$S_{pre}$ // previous solution
*Data*    //includes rooms and patients and Room-Patient-Suitability
*W* // the current window of performing the new optimization
**Output**:
$S_{current}$    // initial solution for current window
**Start**:
1—**for** patient in List of patients from solution
2— Room ← −1 //initialization
3— **if** the patient is delayed (not new)
4—    **if** initial room still has space AND this room is suited for this patient
5—        Room ← previous day solution room
6—    **end if**
7— **end if**
8—    **while** not (Room is suited and has space) AND there is more Rooms
9—      Room ← random (Rooms)
10— **end while**
11— **if** Room not equal to −1
12—    Assign patient to Room.
13—    Set his delay value to zero.
14— **else**    //the case the room is still −1
15—    Assign patient to Room        // if it's delayed, we can use a not suited room.
16—    Set his delay value to one.
17— **end if**
18— **end for**
**End.**

### 5.2.2. Crossover

Crossover attempts to create a new generation from an existing one, encouraging exploitation. In contrast, mutation partially changes existing solutions, encouraging exploration. Genetic algorithms use both crossovers and mutations. The crossover algorithm is presented in Algorithm 5. The input is the entire population, and IN indicates the population percentage where a crossover is performed. Typically, the crossover is applied to the elites, representing the fittest solutions in the generation. The algorithm for each crossover iteration selects two random solutions and generates from each solution a random portion of patients to change their rooms, assigning them to DeltaRooms. In addition, it identifies a random portion of patients to change their delay, assigning them to DeltaDelay. Afterwards, it changes the original selection of two patients and adds the children to the new generation.

---

**Algorithm 5.** The crossover operation for the genetic design.

---

**Input**:
current generation,
IN
**Output**:
new generation
**Start**:
1—Choose a random portion of the generation to apply crossover to.
2—**for** counter IN portion size
3— Choose two parents from the current generation
4— DeltaRooms ← random portion of patients to change their rooms from solution x to solution y.
5— DeltaDelay ← random portion of patients to change their delay from solution x to solution y.
6— Child 1 = change (parent1, parent2, DeltaRooms, DeltaDelay)
7— Child 2 = change (parent2, parent1, DeltaRooms, DeltaDelay)
8— Add child 1 and child 2 to new the generation
9—**end for.**
**End.**

---

5.2.3. Mutation

The mutation pseudocode is presented in Algorithm 6. The input of the algorithm is the individual or solution selected for mutation. The mutation rate indicates how many patients must be changed, while the acceptance rate $ap$ decides when to accept a dominant solution after mutation. Mutation helps escape local optima (i.e., increases exploration).

The mutation algorithm is shown in Algorithm 6. Firstly, it randomly selects the neighborhood type, i.e., type 1 or type 2, and performs the mutation based on the selected individual. Next, the algorithm checks for domination, and it accepts the solution in the case of a dominant solution. In the case of non-domination, the solution is accepted using a predetermined probability (i.e., acceptance rate).

---

**Algorithm 6.** The mutation operation for the genetic design.

---

**Input**:
Solution
Mutation rate: how many patients in the individual to change.
ap: acceptance rate
**Output**:
new Solution with mutated individuals
**Start**:
1—select random neighborhood
2—new-Solution ← neighborhood (Solution, Mutation rate)
3—**if** new-Solution Dominates the current Solution
4— current Solution ← new-Solution
5—**else**
6— Generate a probability to allow bad Solutions
7— **if** generated probability > ap
8— current Solution ← new-Solution
9— **end**
**End.**

---

The neighborhood operation is performed based on neighborhoods 1 and 2, as presented in Algorithms 7 and 8. Neighborhood 1 concentrates on randomly changing the location or room for a random patient, while neighborhood 2 concentrates on randomly changing the delay for a random patient. Both processes are used for the mutation to provide better freedom to the search algorithm.

In Algorithm 7, we present the pseudocode for the mutation operator. It accepts the mutation rate and the current solution and randomly selects patients from the patient list and rooms from the room list. If suitability criteria are met, the patient is assigned the room, and the solution is changed accordingly.

**Algorithm 7.** Pseudocode of neighborhood 1 operator used in the mutation.

**Input**:
Mutation rate
Current Solution
**Output**:
new Solution after the change
**Start**
1—**while** Mutation rate
2— patient ← random (current Solution patients)
3— new-room ← random (current Solution rooms)
4— **if** the new-room is suited for this patient
5— set the patients room to the new-room.
6— **end if**
7—**end while**
**End**

**Algorithm 8.** Pseudocode of neighborhood 2 operator used in the mutation.

**Input**:
Mutation rate
Current Solution
Window
**Output**:
new Solution after the change
**Start**:
1—**while** Mutation rate
2— patient ← random (current individual patients)
3— new-delay ← random (1 ← 0)
4— **if** the new-delay + day is in the patients staying range
5— set the patients delay to the new-delay.
6— **end if**
7—**end while**
**End.**

5.2.4. Evaluation Metrics

This sub-section provides the evaluation metrics used for the formulated approach:

(1)　Set coverage:

This metric compares the Pareto sets $P_{s1}$ and $P_{s2}$ as given in Equation (2).

$$c(P_{s1}, P_{s2}) = \frac{|\{y \in P_{s2} \mid \exists x \in P_{s1} : x > y\}|}{|P_{s2}|} \tag{2}$$

It indicates the number of solutions in $P_{s2}$ dominated by solutions in $P_{s1}$. Next, normalization is performed using the total number of solutions in $P_{s2}$.

(2)　The HV-metric is widely used for evolutionary multi-objective optimization to evaluate algorithm search performance. It computes the volume of the dominated portion of the objective space relative to the least desirable solution (reference point); this region is the union of the hypercube whose diagonal is the distance between the reference point and a solution x from the Pareto set PS. Higher values of this measure indicate more desirable solutions. HV is expressed in Equation (3).

$$HV = \text{volume} \left( U_{x \in P_s} \text{ HyperCube } (x) \right) \tag{3}$$

**6. Experimental Works and Evaluation**

In order to evaluate our developed window-based multi-objective optimization for solving the dynamic patient admission scheduling problem, we used the data provided

by [10]. The data are described by the entity-relationship diagram depicted in Figure 2 The data includes seven entities: patient, treatment, specialty, department, room, feature, and operating room slot (OR Slot). The attributes of each entity are provided in the figure. Detailed descriptions are provided in [10].

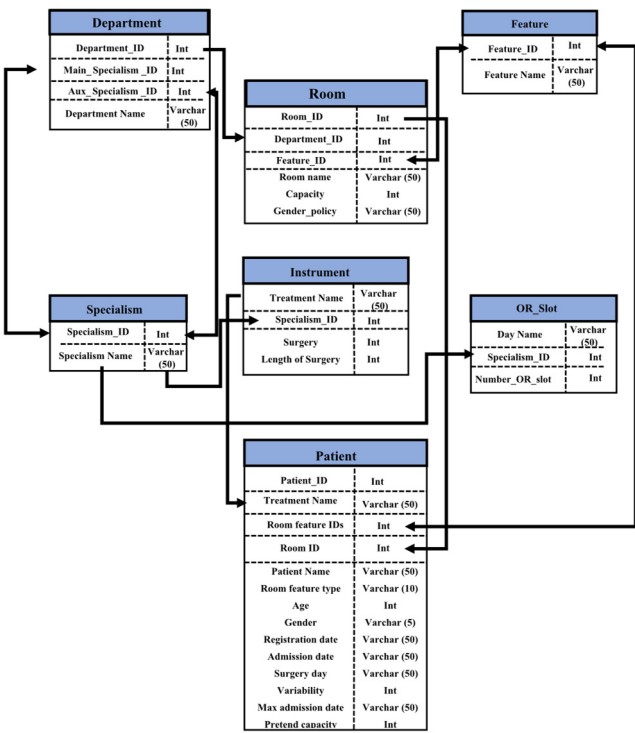

**Figure 2.** Entity relationship diagram (ERD) for describing the data attributes of the data.

We compared the proposed approach with two benchmarks, namely, simulated annealing (SA) and particle swarm optimization (PSO),

(1) Simulated Annealing (SA): the solutions move with their neighborhood area
(2) Particle Swarm Optimization (PSO): the particles (solutions) simultaneously move towards the global and local best particles.

Table 1 provides the weights of the soft constraints that are considered for the single objective algorithms.

**Table 1.** The weights that are assigned to violating soft constraints for the single objective optimization benchmark according to [10].

| Soft Constraints | Corresponding Weight |
|---|---|
| Gender constraint | 5.0 |
| Mandatory Room suitability | 5.0 |
| Age constraint | 10.0 |
| Preferred Room suitability | 2.0 |
| Room category matching | 0.8 |
| The number of transfers | 11.0 |

In addition, we used three multi-objective optimization benchmarks:

(1) MOPSO: the particles simultaneously move towards the best local and the Leader (particle selected from the repository);
(2) ODPSO: it enables objective decomposition when exploration begins;
(3) NSGA II: is a genetic algorithm that selects non-dominated solutions based on crowding distance;

(4)    NSGA III: it is similar to NSGA II but differs concerning the selection stage depending on reference points.

Table 2 lists several parameters: C1, C2 denote common PSO constants; nRep denotes the size of the MOPSO repository; nCrossover denotes the number of individuals in a crossover operation; nMutation denotes the number of individuals in a mutation operation; Mutation Rate denotes the change percentage for one individual.

**Table 2.** The configurations that were used for generating the results of the comparisons.

| Methods | C1 | C2 | nRep | nCrossover | nMutation | Mutation Rate |
|---------|-----|-----|------|-------------|------------|----------------|
| PSO | 10 | 5 | - | - | - | - |
| MOPSO/ODPSO | 10 | 5 | 100 | - | - | - |
| NSGA II | - | - | - | All generation | All generation | 1% |
| NSGA III | - | - | - | All generation | All generation | 1% |

Two evaluations were used: day-based triggering (with the same parameters as [10]) and window-based triggering. Furthermore, the designed genetic operators, crossover and mutation, were used for NSGA-II and NSGA-III. Additionally, particle swarm optimization variants comprising traditional PSO, PSO with objective decomposition (ODPSO), and MOPSO were used for evaluation.

Every algorithm was tested twice, using the approach presented by [10] and then the Window-based approach designated WB. In addition, we compared all genetic and particle swarm optimization variants with the approach specified by [10]. For visualization, we provide the domination adjacency matrix (based on set coverage) in Figure 3. It is evident that the NSGA-III window dominated most benchmarks; its domination over all algorithms was 100%, except PSO-WB (57%).

**Figure 3.** Set coverage of our developed WB approach and its comparison with the benchmarks.

The results indicate that the proposed window-based optimization and genetic and swarm operators had outstanding performance. The second algorithm, PSO-WB, dominated all algorithms with 100% coverage, except for PSO-day.

Hyper-volume was also determined for all algorithms, as depicted in Figure 4. It is evident that the proposed NSGA-III-WB and day provided the highest hyper-volume, indicating NSGA-III-WB domination and choice flexibility.

In order to elaborate on the behavior of all window-based algorithms, we present the superior NSGA-III-WB approach and its convergence curves. As shown in Figure 4, when the optimization was conducted, the cost started from an initial value and decreased until it reached the final optimal value after 150 to 200 iterations. We observed convergence behavior for all algorithms. In addition, we wish to emphasize that the decision

flexibility in the window-based optimization enables patient changes to be made within a time window instead of providing greedy decisions based on the current day when the optimization is performed. Hence, this algorithm is superior to day-by-day optimization. For more elaboration, we present in Figure 5, the convergence curve of NSGA-II in some of the optimization days which show that the algorithm has accomplished an acceptable convergence in all of them.

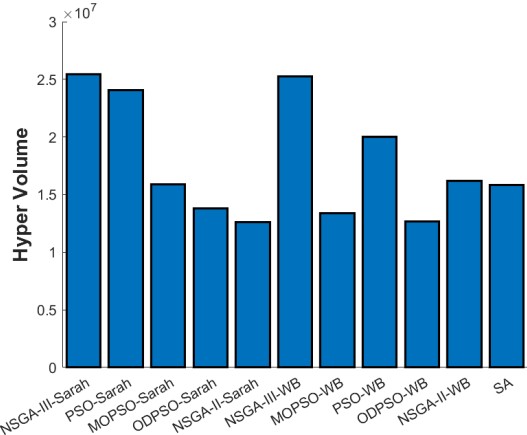

**Figure 4.** Hyper-volume of our developed algorithm and its comparison with the benchmarks.

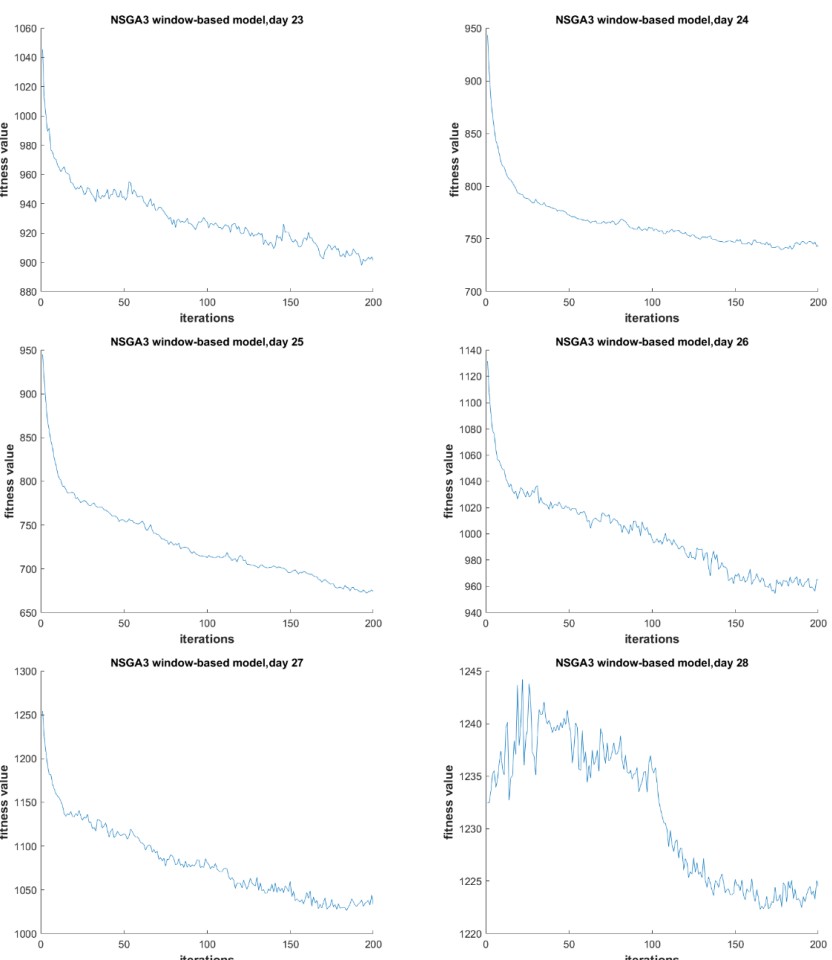

**Figure 5.** The convergence curve of NSGA-III of some of the optimization days (from day 23 until 28).

## 7. Conclusions and Future Work

PAS is a challenging combinatorial optimization problem with a high number of soft constraints. The traditional approach to this problem involved the use of a single objective meta-heuristic optimization based on the weighted sum of the soft constraints; however, this approach causes local minima traps due to the non-convexity of the problem. Another challenging issue facing optimal scheduling is the dynamism that is introduced by the arrival new patients and emergency cases. This article proposes using the Pareto concept based on multi- and many-objective optimization for soft constraints to handle non-convexity and provide more flexibility to decision-makers. At the same time, the article proposes using a window to deal with dynamic aspects and avoid sub-optimality caused by greedy daily rescheduling. The article evaluated two developments based on evolutionary and swarm optimization algorithms and compared them with the traditional single objective day-by-day optimization technique. The novelty is illustrated by the Pareto-based representation of the solutions, which avoid the sub-optimality caused by weighted averages, and window-based scheduling, which avoids the greedy behavior of the day-by-day approach. The findings demonstrate the superiority of the two concepts (Pareto and window) and the overall superiority of joint models. Using many- and window-based optimization produced the best results; the combination of NSGA-III and the window approach was superior to the previous models and benchmark. However, increasing the number of objectives could affect the convergence. To further study this, future work could incorporate more many-objective criteria within the reference point-based selection by NSGA-III.

**Author Contributions:** Conceptualisation, A.N.M.; formal analysis, M.N.M.K. All authors have read and agreed to the published version of the manuscript.

**Funding:** This research received no external funding.

**Institutional Review Board Statement:** Not applicable.

**Informed Consent Statement:** Not applicable.

**Data Availability Statement:** All data has been present in main text.

**Conflicts of Interest:** The authors declare no conflict of interest.

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
