# Peer review of "Window-Based Multi-Objective Optimization for Dynamic Patient Scheduling with Problem-Specific Operators"

_computers, doi:10.3390/computers11050063_

Round 1

Reviewer 1 Report

This paper introduces the Pareto concept based on multi-objective optimisation for soft-constraints to handle non-convexity and provide more flexibility to decision-makers. The proposed method has evaluated by the evolutionary and swarm optimisation algorithms and compared with the traditional single objective day-by-day optimisation technique. Overall, the proposed study is interesting and well written. My opinions and comments are as follows,

  1. In the abstract, what is the “ NP-hard combinatorial optimization problem”?
  2. Figure 2 is not clear.
  3. Please add the list of acronyms.
  4. The authors should further enlarge the Literature Survey with current work about optimization methods to improve the research background, for example: Robust Model Predictive Control Paradigm for Automatic Voltage Regulators against Uncertainty Based on Optimization Algorithms; Effective multi-sensor data fusion for chatter detection in milling process.
  5. More remarks showing the difficulties, novelty and efficiency of the proposed method would be helpful.
  6. Is there any shortcoming of the developed framework? The authors are suggested to make some discussions.
  7. The authors may propose some interesting problems as future work in the conclusion.

Author Response

This paper introduces the Pareto concept based on multi-objective optimisation for soft-constraints to handle non-convexity and provide more flexibility to decision-makers. The proposed method has evaluated by the evolutionary and swarm optimisation algorithms and compared with the traditional single objective day-by-day optimisation technique. Overall, the proposed study is interesting and well written. My opinions and comments are as follows,

  1. In the abstract, what is the “ NP-hard combinatorial optimization problem”?

Response Thank you so much for your suggestion. nondeterministic polynomial time (NP), check line 9.

  1. Figure 2 is not clear.

Response The figure has been updated

  1. Please add the list of acronyms.

Response a list of acronyms is added

  1. The authors should further enlarge the Literature Survey with current work about optimization methods to improve the research background, for example: Robust Model Predictive Control Paradigm for Automatic Voltage Regulators against Uncertainty Based on Optimization Algorithms; Effective multi-sensor data fusion for chatter detection in milling process.

Response

The references are added to the introduction to show the importance of optimization because it is not related to the PAS problem cheek line 40.

  1. More remarks showing the difficulties, novelty and efficiency of the proposed method would be helpful.

Response

The novelty is summarized by Pareto based representation to the solutions to avoid sub-optimality caused by weighted average and window-based scheduling instead of day by day to avoid the greedy behavior, cheek line 428-430.

  1. Is there any shortcoming of the developed framework? The authors are suggested to make some discussions.

Response

However, increasing the objectives could affect the convergence. In order to resolve this, future work is to incorporate more many-objective criteria added to the reference point based selection by NSGA-III, check line 433-436.

  1. The authors may propose some interesting problems as future work in the conclusion.

Response

future work is to incorporate more many-objective criteria added to the reference point based selection by NSGA-III.

Reviewer 2 Report

The paper addresses the patient admission scheduling (PAS) problem, a complex combinatorial optimization problem. A multi-objective optimization technique is proposed to solve the dynamic variant of PAS.

The introduction and state of the art introduce the problem in an adequate manner and discusses several systems that have already dealt with the patient admission scheduling problem.

However, in the opinion of this reviewer, the article leaves out one of the techniques that is currently achieving the best results in terms of both speed and quality of results: Reinforcement Learning applied to combinatorial optimization. I suggest the authors to include a paragraph commenting on the techniques based on Machine Learning with Reinforcement Learning, indicating that this paper focuses only on the classical optimization techniques based on heuristics and approximation. For this same reason I also suggest including in the "literature survey" section a reference to the paper "Reinforcement learning for combinatorial optimization: A survey" (https://doi.org/10.1016/j.cor.2021.105400).

The style of the pseudocode appearing in the article should be standardized. In some tables, capital letters are used for reserved words, while in others, lowercase and bold letters are used. Please choose a common format for all pseudocode tables.

In addition to the above, there are small errors in the paper that should be corrected:

  • On line 25, reference [21] appears referenced 2 times.
  • In table 2, line 11 of the pseudocode starts with ":". Is that correct?

Author Response

The paper addresses the patient admission scheduling (PAS) problem, a complex combinatorial optimization problem. A multi-objective optimization technique is proposed to solve the dynamic variant of PAS.

The introduction and state of the art introduce the problem in an adequate manner and discusses several systems that have already dealt with the patient admission scheduling problem.

However, in the opinion of this reviewer, the article leaves out one of the techniques that is currently achieving the best results in terms of both speed and quality of results: Reinforcement Learning applied to combinatorial optimization.

1- I suggest the authors to include a paragraph commenting on the techniques based on Machine Learning with Reinforcement Learning, indicating that this paper focuses only on the classical optimization techniques based on heuristics and approximation. For this same reason I also suggest including in the "literature survey" section a reference to the paper "Reinforcement learning for combinatorial optimization: A survey" (https://doi.org/10.1016/j.cor.2021.105400).

Response

Third direction is to use machine learning approaches based on reinforcement learning where the decisions-states are used to train single/multi agent for reaching optimal policy [7]. The difference between meta-heuristic searching and reinforcemnet learning is that the latter stores knowledge of state-decisions-reward and use it for update the knowledge in incremetal way. The formulation of state, action or decision, and reward is a key factor in the performance changing. On the other side, meta-heuristic serching has less incremetal nature but it is simpler, check line 53-59

2- The style of the pseudocode appearing in the article should be standardized. In some tables, capital letters are used for reserved words, while in others, lowercase and bold letters are used. Please choose a common format for all pseudocode tables.

Response

The pseudocodes have been updated

3- In addition to the above, there are small errors in the paper that should be corrected:

  • On line 25, reference [21] appears referenced 2 times.
  • In table 2, line 11 of the pseudocode starts with ":". Is that correct?

Response

Thank you so much for your comment we delete reference [21] on line 25 and delete “:” in table line 11 of the pseudocode

Round 2

Reviewer 1 Report

The paper has improved in this version. I do not have more questions.